# *ϕ*-Informational Measures: Some Results and Interrelations

**DOI:** 10.3390/e23070911

**Published:** 2021-07-18

**Authors:** Steeve Zozor, Jean-François Bercher

**Affiliations:** 1GIPSA-Lab, CNRS, Grenoble INP, University Grenoble Alpes, 38000 Grenoble, France; 2CNRS, LIGM, University Gustave Eiffel, 77454 Marne-la-Vallée, France; jean-francois.bercher@esiee.fr

**Keywords:** *ϕ*-entropy, state-dependent *ϕ*-entropy, (inverse) maximum *ϕ*-entropy problem, *ϕ*-escort distributions, *ϕ*-Fisher information, *ϕ*-moments, generalized Cramér–Rao inequality, *ϕ*-heat equation, generalized de Bruijn

## Abstract

In this paper, we focus on extended informational measures based on a convex function ϕ: entropies, extended Fisher information, and generalized moments. Both the generalization of the Fisher information and the moments rely on the definition of an escort distribution linked to the (entropic) functional ϕ. We revisit the usual maximum entropy principle—more precisely its inverse problem, starting from the distribution and constraints, which leads to the introduction of state-dependent ϕ-entropies. Then, we examine interrelations between the extended informational measures and generalize relationships such the Cramér–Rao inequality and the de Bruijn identity in this broader context. In this particular framework, the maximum entropy distributions play a central role. Of course, all the results derived in the paper include the usual ones as special cases.

## 1. Introduction

Since the pioneer works of von Neumann [1], Shannon [2], Boltzmann, Maxwell, Planck, and Gibbs [3,4,5,6,7,8,9], many investigations were devoted to the generalization of the so-called Shannon entropy and its associated measures [10,11,12,13,14,15,16,17,18,19,20,21,22]. If the Shannon measures are compelling, especially in the communication domain, for compression purposes, many generalizations proposed later on have also showed promising interpretations and applications (Panter–Dite formula in quantification where the Rényi or Havrda–Charvát entropy emerges [23,24,25], encoding penalizing long codewords where the Rényi entropy appears [26,27], for instance). The great majority of the extended entropies found in the literature belongs to a very general class of entropic measures called (h,ϕ)-entropies [13,19,20,28,29,30]. Such a general class (or more precisely the subclass of ϕ-entropies) can be traced back to the work of Burbea and Rao [28]. They offer not only a general framework to study general properties shared by special entropies, but they also offer many potential applications as described for instance in [30]. Note that if a large amount of work deals with divergences, entropies occur as special cases when one takes a uniform reference measure.

In the framework of these generalized entropies, the so-called maximum entropy principle takes a special place. This principle, advocated by Jaynes, states that the statistical distribution that describes a system in equilibrium maximizes the entropy while satisfying the system’s physical constraints (e.g., the center of mass and energy) [31,32,33,34,35]. In other words, it is the less informative law given the constraints of the system. In the Bayesian approach, dealing with the stochastic modeling of a parameter, such a principle (or a minimum divergence principle) is often used to choose a prior distribution for the parameter [22,36,37,38,39]. It also finds its counterpart in communication, clustering, pattern recognition, problems, among many others [32,33,40,41,42,43]. In statistics, some goodness-of-fit tests are based on entropic criteria derived from the same idea of constrained maximal entropic law [44,45,46,47,48,49]. The principle behind such entropic tests lies in the Bregman divergence, measuring a kind of distance between probability distributions, i.e., the empirical distribution given by data and the distribution we assume for the data (reference). It appears that if the empirical distribution and the reference share the same moments, and if the latter is of maximum entropy with these moments as constraints, the divergence reduces to a difference of entropy. In a large number of works using the maximum entropy principle, the entropy used is the Shannon entropy. However, if for some reason, a generalized entropy is considered, the approach used in the Shannon case does not fundamentally change [50,51,52,53].

One can consider the inverse problem which consists in finding the moment constraints leading to the observed distribution as a maximal entropy distribution [50]. Kesavan and Kapur also envisaged a second inverse problem, where both the distribution and the moments are given. The question is thus to determine the entropy so that the distribution is its maximizer. As a matter of fact, dealing with the Shannon entropy, whatever the constraints considered, the maximum entropy distribution falls in the exponential family [33,34,52,54]. Remind that the exponential family is the set of parametric densities (with respect to a measure μ independent on the parameter) of the form $p\left(x\right)=C\left(\theta \right)h\left(x\right)\exp(R{\left(\theta \right)}^{t}S\left(x\right))$ where S(x) is the sufficient statistics [39,55,56,57,58,59,60]. When R(θ)=θ, the family is said to be natural and Z(θ)=1/C(θ) is the partition function, the log-partition function φ(θ)=logZ(θ) being the cumulants generating function. Now, resolving the maximum entropy problem given later on by Equation (Equation 6) in the context of the Shannon entropy, it appears indeed that the maximum entropy distribution falls in the natural exponential family where the sufficient statistics is given by the moment constraints. Considering more general entropies allows to escape from this limitation. Moreover, if the Shannon entropy (or the Gibbs entropy in physics) is well adapted to the study of systems in the equilibrium (or in the thermodynamic limit), extended entropies allow a finer description of systems out of equilibrium [17,61,62,63,64,65], exhibiting their importance. While the problem was considered mainly in the discrete setting by Kesavan and Kapur in [50], we will recall it in the general framework of the ϕ-entropies probability densities with respect to any reference measure, and make a further step considering an extended class of these entropies. Resolving the inverse problem can find applications in goodness-of-fit tests for instance, allowing to design entropies adapted to such tests, in the same line as that of the approaches mentioned above [44,45,46,47,48,49].

While the entropy is a widely used tool for quantifying information (or uncertainty) attached to a random variable or to a probability distribution, other quantities are used as well, such as the moments of the variable (giving information, for instance, on center of mass, dispersion, skewness, or impulsive character), or the Fisher information. In particular, the Fisher information appears in the context of estimation [66,67], in Bayesian inference through the Jeffreys prior [39,68], but also for complex physical systems descriptions [67,69,70,71,72,73].

Although coming from different worlds (information theory and communication, estimation, statistics, and physics), these informational quantities are linked by well-known relations such as the Cramér–Rao inequality, the de Bruijn identity, and the Stam inequality [34,74,75,76]. These relationships have been proved very useful in various areas, for instance, in communications [34,74,75], in estimation [66], or in physics [77,78], among others. When generalized entropies are considered, it is natural to question the other informational measures’ generalization and the associated identities or inequalities. This question gave birth to a large amount of work and is still an active field of research [28,79,80,81,82,83,84,85,86,87,88,89,90]. For instance, the Cramér–Rao inequality is very important as it gives the ultimate precision in terms of mean square error of an estimator of a parameter (i.e., the minimal error we can achieve). However, there is no reason for choosing a quadratic error in general. This choice is often made as it allows to simplify algebra or to derive estimators quite easily (e.g., of minimum mean square error). One may wish to choose other error criteria (mean of another norm of the error) and/or to stress parts of the distribution of the data in the mathematical average. It is thus of high interest to be able to derive Cramér–Rao inequalities in a context as broad as possible.

In this paper, we show that it is possible to build a whole framework, which associates a target maximum entropy distribution to generalized entropies, generalized moments, and generalized Fisher information. In this setting, we derive generalized inequalities and identities relating these quantities, which are all linked in some sense to the maximum entropy distribution.

The paper is organized as follows. In Section 2, we recall the definition of the generalized ϕ-entropy. Thus, we come back to the maximum entropy problem in this general settings. Following the sketch of [50], we present a sufficient condition linking the entropic functional and the maximizing distribution, allowing to both solve the direct and the inverse problems. When the sufficient conditions linking the entropic function and the distribution cannot be satisfied, the problem can be solved by introducing state-dependent generalized entropies, which is the purpose of Section 3. In Section 4, we introduce informational quantities associated to the generalized entropies of the previous sections, such as a generalized escort distribution, generalized moments, and generalized Fisher information. These generalized informational quantities allow to extend the usual informational relations such as the Cramér–Rao inequality, relations precisely saturated (or valid) for the generalized maximum entropy distribution. Finally, in Section 5, we show that the extended quantities allows to obtain an extended de Bruijn identity, provided the distribution follows a nonlinear heat equation. Some examples of ϕ-entropies solving the inverse maximum entropy problem are provided in a short series of appendices, showing, in particular, that the usual quantities are recovered as particular cases (Gaussian distribution, Shannon entropy, Fisher information, and variance).

In the following, we will define a series of generalized information quantities relative to a probability density defined with respect to a given reference measure μ (e.g., the Lebesgue measure when dealing with continuous random variables, discrete measure for discrete-state random variables, etc.). Therefore, rigorously, all these quantities depend on the particular choice of this reference measure. However, for simplicity, we will omit to mention this dependence in the notations along the paper.

## 2. *ϕ*-Entropies—Direct and Inverse Maximum Entropy Problems

The direct problem, i.e., finding the probability distribution of maximum entropy given moments constraints, is a common problem and can find application, for instance, in the Bayesian framework, searching for prior probability distribution as less informative as possible, given some moments [22,36,37,38,39]. It also finds many other applications, as mentioned in the introduction.

Let us first recall the definition of the generalized ϕ-entropies introduced by Csiszàr in terms of divergence, and by Burbea and Rao in terms of entropy:

**Definition** **1**(ϕ-entropy [28])**.**
*Let $\phi :Y\subseteq R_{+}\mapsto R$ be a convex function defined on a convex set Y. Then, if f is a probability distribution defined with respect to a general measure μ on a set $X\subseteq R^{d}$ such that f(X)⊆Y, when this quantity exists,*
(1)$$H_{\phi }\left[f\right]=-\int_{X}\phi \left(f\left(x\right)\right)\;d\mu \left(x\right)$$
*is the ϕ-entropy of f.*

The (h,ϕ)-entropy is defined by $H_{\left(h,\phi \right)}\left[f\right]=h\left(H_{\phi }\left[f\right]\right)$ where *h* is a nondecreasing function. The definition is extended by allowing ϕ to be concave, together with *h* nonincreasing [13,19,20,29,30]. If, additionally, *h* is concave, then the entropy functional $H_{\left(h,\phi \right)}\left[f\right]$ is concave.

As we are interested in the maximum entropy problem, and because *h* is monotone, we can restrict our study to the ϕ-entropies. Additionally, we will assume that ϕ is *strictly* convex and *differentiable*.

A related quantity is the Bregman divergence associated with convex function ϕ:

**Definition** **2**(Bregman divergence and functional Bregman divergence [22,91])**.**
*With the same assumptions as in Definition 1, the Bregman divergence associated with ϕ defined on a convex set Y is given by the function defined on Y×Y,*
(2)$$B_{\phi }\left(y_{1},y_{2}\right)=\phi \left(y_{1}\right)-\phi \left(y_{2}\right)-\phi^{\prime }\left(y_{2}\right)\left(y_{1}-y_{2}\right).$$
*Applied to two functions $f_{i}:X\mapsto Y$, i=1,2, the functional Bregman divergence writes*
(3)$$B_{\phi }\left(f_{1},f_{2}\right)=\int_{X}\;\phi \left(f_{1}\left(x\right)\right)\;d\mu \left(x\right)-\int_{X}\;\phi \left(f_{2}\left(x\right)\right)\;d\mu \left(x\right)-\int_{X}\;\phi^{\prime }\left(f_{2}\left(x\right)\right)\left(f_{1}\left(x\right)-f_{2}\left(x\right)\right)\;d\mu \left(x\right).$$
*A direct consequence of the strict convexity of ϕ is the non-negativity of the (functional) Bregman divergence: $B_{\phi }\left(y_{1},y_{2}\right)\geq 0$ and $B_{\phi }\left(f_{1},f_{2}\right)\geq 0$, with equality if and only if $y_{1}=y_{2}$ and $f_{1}=f_{2}$ almost everywhere respectively.*

From its positivity and equality only when the distributions are (almost everywhere) equal, this divergence defines a kind of distance (it is not, being non-symmetrical) where $f_{2}$ serves as a reference.

More generally, the Bregman divergence is defined for multivariate convex functions, where the derivative is replaced by gradient operator [91]. Extensions for convex function of functions also exist, where the derivative is in the sense of Gâteau [92]. Such general extensions are not useful for our purposes; thus, we restrict ourselves to the above definition where $Y\subseteq R_{+}$.

### 2.1. Maximum Entropy Principle: The Direct Problem

Let us first recall the maximum entropy problem that consists in searching for the distribution maximizing the ϕ-entropy (Equation 1) subject to constraints on some moments $E\left[T_{i}\left(X\right)\right]$ with $T_{i}:R^{d}\mapsto R$, i=1,…,n. This direct problem writes
(4)$$f^{🟉}\;=\underset{f\in D_{T,t}}{\mathrm{argmax}}\left(-\int_{X}\phi \left(f\left(x\right)\right)\;d\mu \left(x\right)\right)$$
with
(5)$$D_{T,t}=\left\{f\geq 0:\;E\left[T_{i}\left(X\right)\right]=t_{i},i=0,\ldots ,n\right\},$$
where $T_{0}\left(x\right)=1$ and $t_{0}=1$ (normalization constraint), $T=\left(T_{0},\ldots ,T_{n}\right),\;t=\left(t_{0},\ldots ,t_{n}\right)$. We are faced to a strictly concave optimization problem (the functional to maximize is concave w.r.t. *f* and the constraints are linear w.r.t. *f*, so that the functional restricted to a linear subspace is still concave). Therefore, the solution exists and is unique. A technique to solve the problem can be to use the classical Lagrange multipliers technique and to solve the Euler–Lagrange equation from the variational problem, but this approach requires mild conditions [50,51,53,93,94,95]. In the following proposition, we recall a sufficient condition relating *f* and ϕ so that *f* is the problem’s solution. This result is proven without the use of the Lagrange technique.

**Proposition** **1**(Maximal ϕ-entropy solution [50]). *Suppose that there exists a probability distribution $f\in D_{T,t}$ satisfying*
(6)$$\phi^{\prime }\left(f(x)\right)=\sum_{i=0}^{n}\lambda_{i}\;T_{i}\left(x\right),$$
*for some $\left(\lambda_{0},\ldots ,\lambda_{n}\right)\in R^{n+1}$. Then, f is the unique solution of the maximal entropy problem *(Equation 4)*.*

**Proof.** Suppose that distribution *f* satisfies Equation (Equation 6) and consider any distribution $g\in D_{T,t}$. The functional Bregman divergence between *f* and *g* writes
$$\begin{array}{ccc} B_{\phi }\left(g,f\right) & = & \int_{X}\phi \left(g\left(x\right)\right)\;d\mu \left(x\right)-\int_{X}\phi \left(f\left(x\right)\right)\;d\mu \left(x\right)-\int_{X}\phi^{\prime }\left(f\left(x\right)\right)\left(g(x)-f(x)\right)\;d\mu \left(x\right)\\ & = & -H_{\phi }\left[g\right]+H_{\phi }\left[f\right]-\sum_{i=0}^{n}\lambda_{i}\int_{X}T_{i}\left(x\right)\left(g(x)-f(x)\right)\;d\mu \left(x\right)\\ & = & H_{\phi }\left[f\right]-H_{\phi }\left[g\right] \end{array}$$
where we used the fact that *g* and *f* are both probability distributions with the same moments $E\left[T_{i}\left(X\right)\right]=t_{i}$. By non-negativity of the Bregman functional divergence, we finally get that
$$H_{\phi }\left[f\right]\geq H_{\phi }\left[g\right]$$
for all distributions *g* with the same moments as *f*, with equality if and only if g=f almost everywhere. In other words, this shows that if *f* satisfies Equation (Equation 6), then it is the desired solution. □

Therefore, given an entropic functional ϕ and moments constraints $T_{i}$, Equation (Equation 6) leads the the maximum entropy distribution $f^{🟉}$. This distribution is parameterized by the $\lambda_{i}$s or, equivalently, by the moments $t_{i}$s.

Note that the reciprocal is not necessarily true, i.e., the maximum entropy distribution does not necessarily satisfies Equation (Equation 6) (i.e., Equation (Equation 6) has not necessarily a solution), as shown, for instance, in [53]. However, the reciprocal is true (i.e., Equation (Equation 6) has a solution) when X is a compact [95] or for any X provided that ϕ is locally bounded on X [96].

### 2.2. Maximum Entropy Principle: The Inverse Problems

As stated in the introduction, two inverse problems can be considered starting from a given distribution *f*. These problems were considered by Kesavan and Kapur in [50] in the discrete framework.

The first inverse problem consists in searching for the adequate moments so that a desired distribution *f* is the maximum entropy distribution of a given ϕ-entropy. This amounts to find functions $T_{i}$ and coefficients $\lambda_{i}$ satisfying Equation (Equation 6). This is not always an easy task, and even not always possible. For instance, it is well known that given moment constraints, the maximum Shannon entropy distribution falls in the exponential family [33,34,52,54]. Therefore, if *f* does not belong to this family, the problem has no solution.

The second inverse problem consists in designing the entropy itself, given a target distribution *f* and given the $T_{i}$s. In other words, given a distribution *f*, Equation (Equation 6) may allow to determine the entropic functional ϕ so that *f* is its maximizer. As mentioned in the introduction, solving this inverse problem can find applications, for instance, in goodness-of-fit tests. In such tests, we would like to determine if data fit a given distribution, say *f*. A natural criterion of fit between an empirical distribution and distribution *f* can be a Bregman divergence, where distribution *f* serves as a reference. As shown in the proof of Proposition 1, when both distributions (empirical, reference) share the same moments and when reference *f* is of maximum entropy subject to these moments, the divergence turns to be a difference of entropy and approaches in the line of [44,45,46,47,48,49] can be applied. Distribution *f* and some moments being given/fixed, the problem is thus to determine the adequate entropy so that *f* is of maximum entropy. This is precisely the inverse problem we deal with now.

As for the direct problem, in the second inverse problem, the solution is parameterized by the $\lambda_{i}$s. Here, required properties on ϕ will shape the domain the $\lambda_{i}$s live in. In particular, ϕ must satisfy:the domain of definition of $\phi^{\prime }$ must include f(X); this will be satisfied by construction;from the strict convexity property of ϕ, $\phi^{\prime }$ must be strictly increasing.

Therefore, because $\phi^{\prime }$ must be strictly increasing, it is clear that solving Equation (Equation 6) requires the following two conditions:(C1)f(x) and $\sum_{i=1}^{n}\lambda_{i}\;T_{i}\left(x\right)$ must have the same variations, i.e., $\sum_{i=0}^{n}\lambda_{i}\;T_{i}\left(x\right)$ is increasing (resp. decreasing, constant) where *f* is increasing (resp. decreasing, constant);(C2)f(x) and $\sum_{i=1}^{n}\lambda_{i}\;T_{i}\left(x\right)$ must have the same level sets,
$$f\left(x_{1}\right)=f\left(x_{2}\right)\;\Leftrightarrow \;\sum_{i=0}^{n}\lambda_{i}\;T_{i}\left(x_{1}\right)\;=\;\sum_{i=0}^{n}\lambda_{i}\;T_{i}\left(x_{2}\right)$$.

For instance, in the univariate case, for one moment constraint,
for $X=R_{+},\;T_{1}\left(x\right)=x$, $\lambda_{1}$ must be negative and f(x) must be decreasing,for $X=R,\;T_{1}\left(x\right)=x^{2}$ or $T_{1}\left(x\right)=\left|x\right|$, $\lambda_{1}$ must be negative and f(x) must be even and unimodal.

Under conditions (C1) and (C2), the solutions of Equation (Equation 6) are given by
(7)$$\phi^{\prime }\left(y\right)=\sum_{i=0}^{n}\lambda_{i}\;T_{i}\;\left(f^{-1}\left(y\right)\right)$$
where $f^{-1}$ can be multivalued. However, even if $f^{-1}$ is multivalued, because of condition (C2), $\phi^{\prime }$ is defined univocally.

Equation (Equation 7) provides thus an effective way to solve the inverse problem. However, there exist situations where there does not exist any set of $\lambda_{i}$s such that conditions (C1)–(C2) are satisfied (e.g., $T_{1}\left(x\right)=x^{2}$ with *f* not even). In such a case, we look for a solution for ϕ in a larger class, i.e., by extending the definition of the ϕ-entropy. This will be the purpose of Section 3. Before focusing on this, let us illustrate the previous result on some examples.

### 2.3. Second Inverse Maximum Entropy Problem: Some Examples

To illustrate the previous subsection, let us analyze briefly three examples: the famous Gaussian distribution (Example 1), the *q*-Gaussian distribution also intensively studied (Example 2), and the arcsine distribution (Example 3). The Gaussian, *q*-Gaussian, and arcsine distributions will serve as a guideline all along the paper. The details of the calculus, together with a deeper study related to the sequel of the paper, are presented in the appendix. Other examples are also given in this appendix. In both three examples, except in the next section, we consider the second-order moment constraint $T_{1}\left(x\right)=x^{2}$.

**Example** **1.**
*Let us consider the well-known Gaussian distribution $f_{X}\left(x\right)=\frac{1}{\sqrt{2\pi }\sigma }\exp\left(-\frac{x^{2}}{2\;\sigma^{2}}\right)$, defined over X=R, and let us search for the ϕ-entropy so that the Gaussian is its maximizer subject to the constraint $T_{1}\left(x\right)=x^{2}$. To satisfy condition (C1) we must have $\lambda_{1}<0$, whereas condition (C2) is always satisfied. Rapid calculations, detailed in Section A.1, and a reparameterization of the $\lambda_{i}$s, give the entropic functional*
ϕ(y)=αylog(y)+βy+γwithα>0.
*This is nothing but the Shannon entropy, up to the scaling factor α, and a shift (to avoid the divergence of the entropy when X is unbounded, one will take γ=0). One thus recovers the long outstanding fact that the Gaussian is the maximum Shannon entropy distribution with the second order moment constraint.*


**Example** **2.**
*Let us consider the q-Gaussian distribution, also known as Tsallis distribution or Student distribution [97,98], $f_{X}\left(x\right)=A_{q}{\left(1-\left(q-1\right)\frac{x^{2}}{\sigma^{2}}\right)}_{+}^{\frac{1}{1-q}}$, where $q>0,\;q\neq 1,\;x_{+}=\max\left(x,0\right)$ and $A_{q}$ is the normalization coefficient, defined over X=R when q<1 or over $X=\left(-\frac{\sigma }{\sqrt{q-1}}\;;\;\frac{\sigma }{\sqrt{q-1}}\right)$ when q>1, and let search for the ϕ-entropy so that the q-Gaussian is its maximizer with the constraint $T_{1}\left(x\right)=x^{2}$. Here, again, condition (C1) is satisfied if and only if $\lambda_{1}<0$, whereas condition (C2) is always satisfied. Rapid calculations detailed in Section A.2 lead to the entropic functional, after a reparameterization of the $\lambda_{i}$s, as,*
$$\phi \left(y\right)=\alpha \;\frac{y^{q}-y}{q-1}+\beta \;y+\gamma \;with\;\alpha >0,$$
*where q is thus an additional parameter of the family. This entropy is nothing but the Havrda–Charvát or Daróczy or Tsallis entropy [12,14,17,97], up to the scaling factor α, and a shift (here also, to avoid the divergence of the entropy when X is unbounded, one will take γ=0). This entropy is also closely related to the Rényi entropy [10] via a one-to-one logarithmic mapping. One recovers the also well known fact that the q-Gaussian is the maximum Havrda–Charvát–Rényi–Tsallis entropy distribution with the second order moment constraint [97]. In the limit case q→1, the distribution $f_{X}$ tends to the Gaussian, whereas the Havrda–Charvát–Rényi–Tsallis entropy tends to the Shannon entropy.*


**Example** **3.**
*Consider the arcsine distribution, $f_{X}\left(x\right)=\frac{1}{\sqrt{s^{2}-\pi^{2}x^{2}}}$ where s>0, defined over $X=\left(-\frac{s}{\pi }\;;\;\frac{s}{\pi }\right)$ and let us determine the entropic functionals ϕ so that $f_{X}$ is the maximum ϕ-entropy distribution subject to the constraint $T_{1}\left(x\right)=x^{2}$. Condition (C2) is always satisfied and now, to fulfill condition (C1) we must impose $\lambda_{1}>0$. Some algebra detailed in Section A.4.1 leads to the entropic functional, after a reparameterization of the $\lambda_{i}$s,*
$$\phi \left(y\right)=\frac{\alpha }{y}+\beta \;y+\gamma \;with\;\alpha >0$$
*(again, to avoid the divergence of the entropy, one can adjust parameter γ). This entropy is unusual and, due to its form, is potentially finite only for densities defined over a bounded support and that are divergent in its boundary (integrable divergence).*


## 3. State-Dependent Entropic Functionals and Minimization Revisited

In order to follow asymmetries of the distribution *f* and address the limitation raised by conditions (C1) and (C2), we propose to allow the entropic functional to also depend on the state variable *x*. Indeed, imagine, for instance, that, for two values $x_{1}\neq x_{2}$, the probability distribution is such that $f\left(x_{1}\right)=f\left(x_{2}\right)$, but, at the same time, $\sum_{i}\lambda_{i}T_{i}\left(x_{1}\right)\neq \sum_{i}\lambda_{i}T_{i}\left(x_{2}\right)$ (for any set of $\lambda_{i}$s). In such a situation, one cannot find a function ϕ so as to satisfy condition (C2). Choosing a functional ϕ depending both on f(x) and *x* can allow to have $\phi \left(x_{1},f\left(x_{1}\right)\right)=\phi \left(x_{2},f\left(x_{2}\right)\right)$ so that we expect it could compensate for the fact that, with a usual entropic functional, condition (C2) cannot be satisfied. In the same vein, imposing a particular form for ϕ(x,f(x)), we also expect to be able to treat the case where condition (C1) cannot be satisfied with a usual entropic functional. Let us first define the hence extended state-dependent ϕ-entropy, before demonstrating that such a extension allows indeed to reach our goal.

**Definition** **3**(State-dependent ϕ-entropy). *Let ϕ:X×Y↦R such that for any $x\in X\subseteq R^{d}$, function ϕ(x,·) is a convex function on the closed convex set $Y\subseteq R_{+}$. Then, if f is a probability distribution defined with respect to a general measure μ on set X and such that f(X)⊆Y,*
(8)$$H_{\phi }\left[f\right]=-\int_{X}\phi \left(x,f\left(x\right)\right)\;d\mu \left(x\right)$$
*will be called state-dependent ϕ-entropy of f. As ϕ(x,·) is convex, then the entropy functional $H_{\phi }\left[f\right]$ is concave. A particular case arises when, for a given partition $\left(X_{1},\ldots ,X_{k}\right)$ of X, functional ϕ writes*
(9)$$\phi \left(x,y\right)=\sum_{l=1}^{k}\phi_{l}\left(y\right){𝟙}_{X_{l}}\left(x\right)$$
*where ${𝟙}_{A}$ denotes the indicator of set A. This functional can be viewed as a “$\left(X_{1},\ldots ,X_{k}\right)$-extension” over X×Y of a multiform function defined on Y, with k branches $\phi_{l}$ and the associated ϕ-entropy will be called $\left(X_{1},\ldots ,X_{k}\right)$-multiform ϕ-entropy.*

As in the previous section, we restrict our study to functionals ϕ(x,y) *strictly convex and differentiable* with respect to *y*.

Following the lines of Section 2, a generalized Bregman divergence can be associated to ϕ under the form $B_{\phi }\left(x,y_{1},y_{2}\right)=\phi \left(x,y_{1}\right)-\phi \left(x,y_{2}\right)-\frac{\partial \phi }{\partial y}\left(x,y_{2}\right)\left(y_{1}-y_{2}\right)$, and a generalized functional Bregman divergence $B_{\phi }\left(f_{1},f_{2}\right)=\int_{X}B_{\phi }\left(x,f_{1}\left(x\right),f_{2}\left(x\right)\right)\;d\mu \left(x\right)$.

With these extended quantities, the direct problem becomes
(10)$$f^{🟉}=\underset{f\in D_{T,t}}{\mathrm{argmax}}\left(-\int_{X}\phi \left(x,f\left(x\right)\right)\;d\mu \left(x\right)\right)$$

Although the entropic functional is now state-dependent, the approach adopted before can be applied here, leading to

**Proposition** **2**(Maximum state-dependent ϕ-entropy solution)**.**
*Suppose that there exists a probability distribution f satisfying*
(11)$$\frac{\partial \phi }{\partial y}\left(x,f(x)\right)=\sum_{i=0}^{n}\lambda_{i}\;T_{i}\left(x\right),$$
*for some $\left(\lambda_{0},\ldots ,\lambda_{n}\right)\in R^{n+1}$, then f is the unique solution of the extended maximum entropy problem *(Equation 10)*.*
*If ϕ is chosen in the $\left(X_{1},\ldots ,X_{k}\right)$-multiform ϕ-entropy class, this sufficient condition writes*
(12)$$\sum_{l=1}^{k}\phi_{l}^{\prime }\left(f(x)\right)\;{𝟙}_{X_{l}}\left(x\right)=\sum_{i=0}^{n}\lambda_{i}\;T_{i}\left(x\right),$$


**Proof.** The proof follows the steps of Proposition 1, using the generalized functional Bregman divergence instead of the usual one. □

Resolving Equation (Equation 11) is not possible in all generality. However, the sufficient condition (Equation 12) can be rewritten as
(13)$$\sum_{l=1}^{k}\left(\phi_{l}^{\prime }\left(f(x)\right)-\sum_{i=0}^{n}\lambda_{i}\;T_{i}\left(x\right)\right)\;{𝟙}_{X_{l}}\left(x\right)=0.$$
Therefore, if there exists (at least) a set of $\lambda_{i}$s such that condition (C1) is satisfied (but not necessarily (C2)), one can always

design a partition $\left(X_{1},\ldots ,X_{k}\right)$ so that (C2) is satisfied *in each $X_{l}$* (at least, such that *f* is either strictly monotonic, or constant, on $X_{l}$) anddetermine $\phi_{l}$ as in Equation (Equation 7) in each $X_{l}$, that is
(14)$$\phi_{l}^{\prime }\left(y\right)=\sum_{i=0}^{n}\lambda_{i}\;T_{i}\left(f_{l}^{-1}\left(y\right)\right)$$
where $f_{l}^{-1}$ is the (possibly multivalued) inverse of *f* on $X_{l}$. By the way, when $X_{l}$ is such that $f_{X}$ is monotonic on it ensures that $f_{l}^{-1}$ is univalued.

In short, in the case where only condition (C1) is satisfied, one can obtain an extended entropic functional of $\left(X_{1},\ldots ,X_{k}\right)$-multiform class so that Equation (Equation 13) provides an effective way to solve the inverse problem in the state-dependent entropic functional context. This is given by Equation (Equation 14).

Note, however, that it still may happen that there is no set of $\lambda_{i}$s allowing to satisfy (C1). In this harder context, the problem remains solvable when the moments are defined as partial moments like $E\left[T_{l,i}\left(X\right){𝟙}_{X_{l}}\left(X\right)\right]=t_{l,i}$, l=1,…,k and $i=1,\ldots ,n_{l}$ and when there exists on $X_{l}$ a set of $\lambda_{l,i}$s such that (C1) and (C2) hold. The solution still writes as in Equation (Equation 14), but where now *n*, the $\lambda_{i}$s and the $T_{i}$s are replaced by $n_{l}$, the $\lambda_{l,i}$s and $T_{l,i}$s, respectively,
(15)$$\phi_{l}^{\prime }\left(y\right)=\sum_{i=0}^{n_{l}}\lambda_{l,i}\;T_{l,i}\left(f_{l}^{-1}\left(y\right)\right)$$

Let us now come back to the arcsine example $f_{X}\left(x\right)=\frac{1}{s^{2}-\pi^{2}x^{2}}$, defined over $X=\left(-\frac{s}{\pi }\;;\;\frac{s}{\pi }\right)$ (Example 3) of the previous section, when now we constraint the first order moment or partial first order moments.

**Example** **4.**
*Let us now consider this arcsine distribution, constrained uniformly by $T_{1}\left(x\right)=x$. Clearly, neither condition (C1) nor condition (C2) can be satisfied. Note that the arcsine distribution is a one-to-one function on each set $X_{-}=\left(-\frac{s}{\pi }\;;\;0\right)$ and $X_{+}=\left[0\;;\;\frac{s}{\pi }\right)$ that partitions X. Therefore, considering multiform entropic functionals with this partition allows to overcome the issue on condition (C2), but that on condition (C1) remains. If we ignore this issue and apply Equation *(Equation 14)*, after a reparameterization of the $\lambda_{i}$s, we obtain ${\tilde{\phi }}_{\pm }\left(y\right)={\tilde{\phi }}_{\pm ,u}\left(sy\right)$ with ${\tilde{\phi }}_{\pm ,u}\left(u\right)=\pm \alpha \left(\sqrt{u^{2}-1}+\arctan\left(\frac{1}{\sqrt{u^{2}-1}}\right)\right)\;{𝟙}_{\left(1\;;\;+\infty \right)}\left(u\right)+\beta \;u+\gamma_{\pm }$ where s is thus an additional parameter of the family. It appears that whereas these functionals are defined for u>1, one can extend them continuously and with a continuous derivative for any u>0 imposing β=0, which finally leads to the family*
$$\begin{array}{ccc} & & {\tilde{\phi }}_{\pm }\left(y\right)={\tilde{\phi }}_{\pm ,u}\left(sy\right)\;with\\ & & {\tilde{\phi }}_{\pm ,u}\left(u\right)=\pm \alpha \left(\sqrt{u^{2}-1}+\arctan\left(\frac{1}{\sqrt{u^{2}-1}}\right)\right)\;{𝟙}_{\left(1\;;\;+\infty \right)}\left(u\right)\;+\;\gamma_{\pm } \end{array}$$
*However, the functional are no more convex (see Section A.4.2 for more details).*


**Example** **5.**
*If now we impose the partial constraint $T_{\pm ,1}\left(x\right)=x{𝟙}_{X_{\pm }}\left(x\right)$, and search for the ϕ-entropy so that $f_{X}$ is the maximizer subject to these constraints, condition (C1) can be now satisfied on each $X_{\pm }$ by imposing the $\pm \lambda_{\pm ,1}$ given Equation *(Equation 15)* to be positive. We then obtain the associated multiform entropic functional, after a reparameterization of the $\lambda_{i}$s, as $\phi_{\pm }\left(y\right)=\phi_{\pm ,u}\left(sy\right)$ with $\phi_{\pm ,u}\left(u\right)=\alpha_{\pm }\left(\sqrt{u^{2}-1}+\arctan\left(\frac{1}{\sqrt{u^{2}-1}}\right)\right)\;{𝟙}_{\left(1\;;\;+\infty \right)}\left(u\right)+\beta \;u+\gamma_{\pm }$ with $\alpha_{\pm }>0$ and where s is thus an additional parameter of the family. In this case, the entropic functionals can be considered for any u>0 by imposing β=0, and one can check that the obtained functions are of class $C^{1}$. This finally leads to the family*
$$\begin{array}{ccc} & & \phi_{\pm }\left(y\right)={\tilde{\phi }}_{\pm ,u}\left(sy\right)\;with\\ & & \phi_{\pm ,u}\left(u\right)=\alpha_{\pm }\left(\sqrt{u^{2}-1}+\arctan\left(\frac{1}{\sqrt{u^{2}-1}}\right)\right)\;{𝟙}_{\left(1\;;\;+\infty \right)}\left(u\right)\;+\;\gamma_{\pm },\;\alpha_{\pm }>0 \end{array}$$
*In addition, remarkably, the entropic functional can be made univalued by choosing $\alpha_{+}=\alpha_{-}$ and $\gamma_{+}=\gamma_{-}$. In fact, such a choice is equivalent to considering the constraint $T_{1}\left(x\right)=\left|x\right|$ which respects the symmetries of the distribution and allows to recover a classical ϕ-entropy (see Section A.4.2 for more details).*


At a first glance, the solutions of Examples 4 and 5 seem to be identical. In fact, they drastically differ. Indeed, let us emphasize that the problem has one constraint in the first case, but two in the second case. The consequence is that four parameters parameterize the first solution $\beta ,\gamma_{\pm }$ and α, while five parameters $\beta ,\gamma_{\pm }$ and $\alpha_{\pm }$ parameterize the second solution. This difference is not insignificant: the first case cannot be viewed as a special case of the second one, because $\alpha_{\pm }$ must be positive, which cannot be possible with only parameter α as ±α rule the ${\tilde{\phi }}_{\pm }$. For the first example, the solution does not lead to a convex function, because this would contradict the required condition (C1) on the parts $X_{\pm }$. Coming back to the direct problem, the “ϕ-like-entropy” defined with $\tilde{\phi }$ is no more concave (indeed, it is no more an entropy in the sense of Definition 1). As such, the maximum ϕ-entropy problem is no more concave: one cannot guarantee the uniqueness and even the existence of a maximum so that there is no guarantee that the arcsine distribution would be a maximizer. Indeed, Equation (Equation 6) coming from the Euler-Lagrange equation (see paragraph previous to Proposition 1), one can just conclude that the arcsine is a critical point (either extremal, or inflection point) of the identified ϕ-like-entropy.

In Section 2 and Section 3, we established general entropies with a given maximizer. In what follows, we will complete the information theoretical setting by introducing generalized escort distributions, generalized moments, and generalized Fisher information associated to the same entropic functional. We will then explore some of their relationships. Indeed, as mentioned in the introduction, the Cramér–Rao inequality is very important as it gives the ultimate precision in terms of mean square error of an estimator of a parameter. Aswe would like to escape from the usual quadratic loss (that has often mathematical motivation but not physical one, and that even can not exist) and/or to stress parts of the distribution of the data so has to penalize for instance large errors depending of the tails of the distribution, it is thus of high interest to be able to derive Cramér–Rao inequalities in a broader framework, which can find natural applications in the estimation domain.

## 4. ϕ-Escort Distribution, (ϕ,α)-Moments,
(ϕ,β)-Fisher Information, Generalized Cramér–Rao Inequalities

In this section, we begin by introducing the above-mentioned informational quantities. We will then show that generalizations of the celebrated Cramér–Rao inequalities hold and link the generalized moments and Fisher information. Furthermore, the lower bound of the inequalities are saturated precisely by maximal ϕ-entropy distributions. To derive such generalizations of this inequality, we thus need to precisely define the above mentioned generalization of the moments and of the Fisher information that will lower bound the moment (e.g., of any estimator of a parameter). The proposed generalizations are based on the notion of escort distribution we first need to introduce.

Escort distributions have been introduced as an operational tool in the context of multifractals [99,100], with interesting connections with the standard thermodynamics [101] and with source coding [26,27]. In our context, we also define (generalized) escort distributions associated with a particular convex function ϕ, and show how they pop up naturally. It is then possible to define generalized moments with respect to these escort distributions. Such distributions were previously introduced dealing with Rényi entropies and took the form $f^{q}$ as we will see later on. When q>1, the effect is to stress the head of the distribution, i.e., to penalize more the errors where the data fall in the head of the distribution. At the opposite, when q<1, the tails of the distributions are stressed. As we will see later on in the proof of the generalized Cramér–Rao inequality, any form as an escort distribution can be chosen. However, as for the usual nonparametric Cramér–Rao inequality, one may wish the inequality to be saturated for the maximum entropy distribution, which fixes the form of the escort distribution as follows.

**Definition** **4**(ϕ-escort)**.**
*Let ϕ:X×Y↦R such that for any $x\in X\subseteq R^{d}$ function ϕ(x,·) is a strictly convex twice differentiable function defined on the closed convex set $Y\subseteq R_{+}$. Then, if f is a probability distribution defined with respect to a general measure μ on a set X such that f(X)⊆Y, and such that*
(16)$$C_{\phi }\left[f\right]\;=\;\int_{X}\frac{d\mu (x)}{\frac{\partial^{2}\phi }{\partial y^{2}}\left(x,f\left(x\right)\right)}\;<\;+\infty$$
*we define by*
(17)$$E_{\phi ,f}\left(x\right)=\frac{1}{C_{\phi }\left[f\right]\;\;\frac{\partial^{2}\phi }{\partial y^{2}}\left(x,f\left(x\right)\right)}$$
*the ϕ-escort density with respect to measure μ, associated to density f.*

Note that from the strict convexity of ϕ with respect to its second argument, this probability density is well defined and is strictly positive. We can note that, with the above definition, the ϕ-escort distribution will tend to stress the parts of the distribution where ϕ(x,f(x)) has a small “curvature.” Moreover, coming back to the previous examples, one can see the following.

**Example** **1** **(cont.).**
*In the context of the Shannon entropy, entropy for which the Gaussian is the maximal entropy law for the second order moment constraint, ϕ(x,y)=ϕ(y)=ylogy, the ϕ-escort density associated to f restricts to density f itself.*


**Example** **2** **(cont.).**
*In the Rényi–Tsallis context, entropy for which the q-Gaussian is the maximal entropy law for the second-order moment constraint $\phi \left(x,y\right)=\phi \left(y\right)=\frac{y^{q}-y}{q-1}$, and $E_{\phi ,f}\propto f^{2-q}$ which recovers the escort distributions used in the Rényi–Tsallis context up to a duality transformation [101].*


**Example** **3** **(cont.).**
*For the entropy that is maximal for the arcsine distribution under the second order moment constraint, $\phi \left(x,y\right)=\phi \left(y\right)=\frac{1}{y}$, and $E_{\phi ,f}\propto f^{3}$ which is nothing more than an escort distributions used in the Rényi–Tsallis context. Indeed, although the arcsine distribution does not fall in the q-Gaussian family, its form is very similar to a q-Gaussian distribution (with q=−1) where the “scaling” parameter would not be related to the exponent q. It is thus not surprising to recover an escort distribution associated to this family.*


**Definition** **5**((α,ϕ)-moments)**.**
*Under the assumptions of Definition 4, with X equipped with a norm ${∥\cdot ∥}_{\chi }$, we define the (α,ϕ)-moment of a random variable X associated to distribution f by*
(18)$$M_{\alpha ,\phi }\left[f;X\right]=\int_{X}{∥x∥}_{\chi }^{\;\alpha }\;E_{\phi ,f}\left(x\right)\;d\mu \left(x\right)$$
*if this quantity exists.*

This definition goes further than the usual definition of variance as a measure of dispersion, both by generalizing the exponent, the norm, and by taking the mean with respect to an escort distribution. Thanks to the escort distribution, one can stress special parts of the distribution (heads, tails, parts where ϕ has a small curvature that is with a small informational content in a sense). Here, again, any escort distribution could have been chosen, but, as pointed out previously, that of the definition allows to saturate the Cramér–Rao inequality we will derive in a while for the maximum entropy distribution. Note that, in the particular case of the Euclidean norm and α=2, the second-order moment statistics are indeed contained in the second-order moments matrix given by the mathematical mean of $XX^{t}$. In such a context, the definition above coincides with the trace of this second order moment matrix and represents the total power of *X*.

This said, for our three examples, we have the following.

**Example** **1** **(cont.).**
*In the context of the Shannon entropy, the (α,ϕ)-moments are the usual moments of ${∥X∥}_{\chi }^{\alpha }$.*


**Example** **2** **(cont.).**
*In the Rényi–Tsallis context the generalized moments introduced in [61,102] are recovered.*


**Example** **3** **(cont.).**
*For $\phi \left(x,y\right)=\phi \left(y\right)=\frac{1}{y}$, one also naturally finds generalized moments with the same form as those introduced in [61,102] (see the items related to the escort distributions).*


The Fisher information’s importance is well known in estimation theory: the estimation error of a parameter is bounded by the inverse of the Fisher information associated with this distribution [34,66]. The Fisher information is also used as a method of inference and understanding in statistical physics and biology, as promoted by Frieden [67] and has been generalized in the Rényi–Tsallis context in a series of papers [81,84,86,87,88,89,103,104]. In the following, we generalize these definitions a step further in our ϕ-entropy context.

**Definition** **6**(Nonparametric (β,ϕ)-Fisher information)**.**
*With the same assumption as in Definition 4, denoting by ${∥\cdot ∥}_{\chi *}$ the dual norm (the norm induced in the dual space that gives here ${∥z∥}_{\chi^{*}}=\underset{{∥x∥}_{\chi }=1}{\sup}z^{t}x$ [105,106]), for any differentiable density f, we define the quantity*
(19)$$I_{\beta ,\phi }\left[f\right]=\int_{X}{∥\frac{\nabla_{x}f\left(x\right)}{E_{\phi ,f}\left(x\right)}∥}_{\chi *}^{\beta }\;E_{\phi ,f}\left(x\right)\;d\mu \left(x\right)$$
*if this quantity exists, as the nonparametric (β,ϕ)-Fisher information of f.*

Note that the Fisher information can be viewed as local, as it is sensitive to the variation of a distribution, rather than to the distribution itself. As for the generalized moments, through the power β other moments for the gradient of *f* than the second one can be considered, so that more or less weight can be put in the variations of the distribution. Moreover, as for the case of generalized moments, any escort distribution could have been chosen, but, again this choice is dictated by our wish to saturate the Cramér–Rao inequality for the maximum entropy distribution.

Note also that when ϕ is state-independent, ϕ(x,y)=ϕ(y), as for the usual Fisher information, this quantity is shift-invariant, i.e., for $g\left(x\right)=f\left(x-x_{0}\right)$ one has $I_{\beta ,\phi }\left[g\right]=I_{\beta ,\phi }\left[f\right]$. This property is unfortunately lost in the state-dependent context. Furthermore, whereas the Fisher information have scaling properties $I\left[a^{-d}f\left(\cdot /a\right)\right]=I\left[f\right]/a^{2}$, this is lost for $I_{\beta ,\phi }$, except when $\phi^{\prime \prime }$ is a power (which corresponds either to the Shannon or Rényi–Tsallis entropy).

**Definition** **7**(Parametric (β,ϕ)-Fisher information). *Let us consider the same assumptions as in Definition 4, and a density f parameterized by $\theta \in \Theta \subseteq R^{m}$ where set *Θ* is equipped with a norm ${∥\cdot ∥}_{\Theta }$ and with the corresponding dual norm denoted ${∥\cdot ∥}_{\Theta *}$. Assume that f is differentiable with respect to θ. We define by*
(20)$$I_{\beta ,\phi }\left[f;\theta \right]=\int_{X}{∥\frac{\nabla_{\theta }f\left(x\right)}{E_{\phi ,f}\left(x\right)}∥}_{\Theta *}^{\beta }\;E_{\phi ,f}\left(x\right)\;d\mu \left(x\right)$$
*as the parametric (β,ϕ)-Fisher information of f.*

Note that, as for the usual Fisher information, when the norms on X and on Θ are the same, the nonparametric and parametric information coincide when θ is a location parameter.

Note that in the classical setting, the information on *X* in the sense of Fisher is given by the so-called Fisher information matrix, which is the mathematical mean of $\nabla f\;\nabla^{t}f$. Taking the trace of the Fisher information matrix, one obtains what is often called Fisher information (without the term “matrix”), which is nothing but the expectation of ${∥\nabla f∥}^{2}$ [58,67,107]. This is in the line of the above definitions. Extending these definitions to obtain a matrix would have been possible by averaging over the ϕ-escort distribution the element-wise power β/2 of matrix $\left(\nabla f\;\nabla^{t}f\right)/E_{\phi ,f}^{2}$, but the trace of this matrix does not coincide anymore with the above definition. Moreover, it is not obvious that it will allow a generalization of the matrix form of the Cramér–Rao inequality we will see in the following. Such a matrix extended Fisher information is left as a perspective.

For our three examples, we have the following.

**Example** **1** **(cont.).**
*In the Shannon entropy context, when the norm is the Euclidean norm and β=2, the nonparametric and parametric information (β,ϕ)-Fisher give the usual nonparametric and parametric Fisher information, respectively.*


**Example** **2** **(cont.).**
*Similarly, in the Rényi–Tsallis context, the generalizations proposed in [87,88,89] are recovered.*


**Example** **3** **(cont.).**
*For $\phi \left(x,y\right)=\phi \left(y\right)=\frac{1}{y}$, one also naturally finds, the generalizations proposed in [87,88,89] (see the items related to the escort distributions).*


We have now the quantities that allow to generalize the Cramér–Rao inequalities as follows.

**Proposition** **3**(Nonparametric (α,ϕ)-Cramér–Rao inequality)**.**
*Assume that a differentiable probability density function with respect to a measure μ, defined on a domain X, admits an (α,ϕ)-moment and an $\left(\alpha^{*},\phi \right)$-Fisher information with α≥1 and $\alpha^{*}$ its Hölder-conjugated, $\frac{1}{\alpha }+\frac{1}{\alpha^{*}}=1$, and that xf(x) vanishes at the boundary of X. Thus, density f satisfies the (α,ϕ) extended Cramér–Rao inequality*
(21)$$M_{\alpha ,\phi }{\left[f;X\right]}^{\frac{1}{\alpha }}\;I_{\alpha^{*}\;,\phi }{\left[f\right]}^{\frac{1}{\alpha^{*}}}\;\geq \;d$$
*When ϕ is state-independent, ϕ(x,y)=ϕ(y), the equality occurs when f is the maximal ϕ entropy distribution subject to the moment constraint $T\left(x\right)={∥x∥}_{\chi }^{\;\alpha }$.*

**Proof.** The approach follows [89], starting from the differentiable probability density *f* (derivative denoted $\nabla_{x}f$), as xf(x) vanishes in the boundaries of *X* from the divergence theorem one has
$$0=\int_{X}\nabla_{x}^{t}\left(xf(x)\right)\;d\mu \left(x\right)=\int_{X}\left(\nabla_{x}^{t}\;x\right)f\left(x\right)\;d\mu \left(x\right)+\int_{X}x^{t}\left(\nabla_{x}f\left(x\right)\right)\;d\mu \left(x\right)$$
Now, for the first term, we use the facts that $\nabla_{x}^{t}x=d$ and that *f* is a density to achieve
$$d=-\int_{X}x^{t}\frac{\nabla_{x}f\left(x\right)}{g(x)}\;g\left(x\right)\;d\mu \left(x\right)$$
for any function *g* non-zero on X. Now, noting that d>0, we obtain from the work in [89] (Lemma 2)
$$\begin{array}{ccc} d & = & \left|\int_{X}x^{t}\left(\frac{\nabla_{x}f\left(x\right)}{g(x)}\right)g\left(x\right)\;d\mu \left(x\right)\right|\\ & \leq & {\left(\int_{X}{∥x∥}_{\chi }^{\;\alpha }\;g\left(x\right)\;d\mu \left(x\right)\right)}^{\frac{1}{\alpha }}{\left(\int_{X}{∥\frac{\nabla_{x}f\left(x\right)}{g(x)}∥}_{\chi *}^{\alpha^{*}}g\left(x\right)\;d\mu \left(x\right)\right)}^{\frac{1}{\alpha^{*}}} \end{array}$$
The proof ends by choosing $g=E_{\phi ,f}$ the ϕ-escort density associated to density *f*. Note now that, again from [89] (Lemma 2), the equality is obtained when
$$\nabla_{x}f\left(x\right)\;\frac{\partial^{2}\phi }{\partial y^{2}}\left(x,f\left(x\right)\right)=\lambda_{1}\nabla_{x}{∥x∥}_{\chi }^{\;\alpha }$$
where $\lambda_{1}$ is a negative constant. Consider now the case where ϕ(x,y)=ϕ(y) is state-independent. Thus, $\nabla_{x}f\left(x\right)\;\frac{\partial^{2}\phi }{\partial y^{2}}\left(x,f\left(x\right)\right)=\nabla_{x}\phi^{\prime }\left(f\left(x\right)\right)$, that gives
$$\phi^{\prime }\left(f\left(x\right)\right)=\lambda_{0}+\lambda_{1}{∥x∥}_{\chi }^{\;\alpha }$$
This last equation has precisely the form Equation (Equation 6) of Proposition 1. □

Analyzing minutely the proof, it is clear that both in the generalized moments and the generalized Fisher information, any escort distribution *g* can be chosen (being identical for both quantities), including the probability distribution itself. The saturation will be achieved for the distribution *f* satisfying $\nabla_{x}f\left(x\right)\;g\left(x\right)=\lambda_{1}\nabla_{x}{∥x∥}_{\chi }^{\;\alpha }$, but the ϕ-escort distribution Definition 4 is the only choice which allows to recover maximal ϕ-entropy as the saturating distribution; of course with the same ϕ as in the escort distribution, and with the moment constraint similar to that of the inequality but averaged over the distribution itself.

An obvious consequence of the proposition is that the probability density that minimizes the $\left(\alpha^{*},\phi \right)$-Fisher information subject to the moment constraint $T\left(x\right)={∥x∥}_{X}^{\alpha }$ coincides with the maximal ϕ-entropy distribution subject to the same moment constraint.

In the problem of estimation, the purpose is to determine a function $\hat{\theta }\left(x\right)$ in order to estimate an unknown parameter θ. In such a context, the Cramér–Rao inequality allows to lower bound the variance of the estimator thanks to the parametric Fisher information. The idea is thus to extend this to bound any α order mean error using our generalized Fisher information.

**Proposition** **4**(Parametric (α,ϕ)-Cramér–Rao inequality)**.**
*Let f be a probability density function with respect to a general measure μ defined over a set X, where f is parameterized by a parameter $\theta \in \Theta \subseteq R^{m}$, and satisfies the conditions of Definition 7. Assume that both μ and X do not depend on θ, that f is a jointly measurable function of x and θ which is integrable with respect to x and absolutely continuous with respect to θ, and that the derivatives of f with respect to each component of θ are locally integrable. Thus, for any estimator $\hat{\theta }\left(X\right)$ of θ that does not depend on θ, we have*
(22)$$M_{\alpha ,\phi }\;{\left[f;\hat{\theta }\left(X\right)-\theta \right]}^{\frac{1}{\alpha }}\;I_{\alpha^{*}\;,\phi }{\left[f;\theta \right]}^{\frac{1}{\alpha^{*}}}\;\geq \;\left|\;m\;+\;\nabla_{\theta }^{t}\;b\left(\theta \right)\;\right|$$
*where*
(23)$$b\left(\theta \right)=E\left[\hat{\theta }\left(X\right)-\theta \right]$$
*is the bias of the estimator and α and $\alpha^{*}$ are Hölder conjugated. When ϕ is state-independent, ϕ(x,y)=ϕ(y), the equality occurs when f is the maximal ϕ entropy distribution subject to the moment constraint $T\left(x\right)={∥\hat{\theta }\left(x\right)-\theta ∥}_{\Theta }^{\;\alpha }$.*

**Proof.** The proof follows again that of [89], and starts by evaluating the divergence of the bias. The regularity conditions in the statement of the theorem enable to interchange integration with respect to *x* and differentiation with respect to θ, so that
$$\nabla_{\theta }^{t}\;b\left(\theta \right)\;=\;\int_{X}\left(\nabla_{\theta }^{t}\;\hat{\theta }\left(x\right)\;-\;\nabla_{\theta }^{t}\;\theta \right)f\left(x\right)\;d\mu \left(x\right)\;+\;\int_{X}{\left(\hat{\theta }\left(x\right)-\theta \right)}^{t}\nabla_{\theta }f\left(x\right)\;d\mu \left(x\right)$$
Note then that $\nabla_{\theta }^{t}\;\theta \;=\;m$ and that $\hat{\theta }$ being independent on θ, one has $\nabla_{\theta }^{t}\;\hat{\theta }\left(x\right)\;=\;0$. Thus, *f* being a probability density, the equality becomes
$$m+\nabla_{\theta }^{t}\;b\left(\theta \right)=\int_{X}{\left(\hat{\theta }\left(x\right)-\theta \right)}^{t}\frac{\nabla_{\theta }f\left(x\right)}{g(x)}\;g\left(x\right)\;d\mu \left(x\right)$$
for any density *g* non-zero on X. The proof ends with the very same steps that in Proposition 4 using [89] (Lemma 2). □

In the classical setting, in the multivariate context (m>1), the Cramér–Rao inequality takes a matrix form, stating that the difference of the second order moment matrix of the estimation error of an estimator with the inverse Fisher information matrix is positive definite [34,58,66,67,108,109]. Several scalar forms can be derived, for instance by taking the determinant, the trace, and/or by mean of trace [58,66,67,108] or determinant/trace inequalities [110]. Typically, by mean of the trace, the scalar equivalent of the above results are recovered. Conversely, extending our result in a matrix context is not immediate and left as a perspective.

For our three examples, Propositions 3 and 4 lead to what follows.

**Example** **1** **(cont.).**
*The usual parametric and nonparametric Cramér–Rao inequality are recovered in the usual Shannon context ϕ(x,y)=ylogy, using the Euclidean norm and α=2. The bound in the nonparametric context is saturated for the maximal entropy law, namely, the Gaussian.*


**Example** **2** **(cont.).**
*In the Rényi–Tsallis context, the generalizations proposed in [87,88,89] are recovered and, again, when α=2, the bound is saturated in the nonparametric context for the q-Gaussian, maximal entropy law under the second order moment constraint.*


**Example** **3** **(cont.).**
*For $\phi \left(x,y\right)=\phi \left(y\right)=\frac{1}{y}$, again, one finds inequalities with the same form as those of the generalizations proposed in [87,88,89] (see the items related to the escort distributions).*


Beyond the mathematical aspect of these relations, they may have great interest to assess an estimator when the usual variance/mean square error does not exist. Moreover, the escort distribution is also a way to emphasize some part of a distribution. For instance, in the Rényi–Tsallis context, one can see that in $f^{q}$ either the tails or the head of the distribution are emphasized. Playing with *q* is a way to penalize either the tails, or the head of the distribution in the estimation process.

## 5. ϕ-Heat Equation and Extended de Bruijn Identity

An important relation connecting the Shannon entropy *H*, coming from the “information world”, with the Fisher information *I*, living in the “estimation world”, is given by the de Bruijn identity and it is closely linked to the Gaussian distribution. Considering a noisy random variable $Y_{\theta }=X+\sqrt{\theta }N$ where *N* is a zero-mean *d*-dimensional standard Gaussian random vector and *X* a *d*-dimensional random vector independent of *N*, and of support independent on parameter θ, then
$$\frac{d}{d\theta }H\left[f_{Y_{\theta }}\right]=\frac{1}{2}I\left[f_{Y_{\theta }}\right]$$
where $f_{Y_{\theta }}$ stands for the probability distribution of $Y_{\theta }$. This identity is a critical ingredient in proving the entropy power and Stam inequalities [34]. The de Bruijn identity has applications in communication by characterizing a channel face to noise [34,76,111,112] or in mismatch estimation [113]. It is involved in the Entropy Power Inequality, which itself is involved in an informational proof of the central limit theorem [114,115,116]. Extending the de Bruijn identity is thus of great interest as, for instance, it may allow to characterize more general communication channels in the same line than that in [117] or for non-additive noise or to give rise to generalized central limit theorem [115,116].

The starting point to establish the de Bruijn identity is the heat equation satisfied by the probability distribution $f_{Y_{\theta }}$, $\frac{\partial f}{\partial \theta }=\frac{1}{2}\Delta f$, where Δ stands for the Laplacian operator [118].

Let us consider probability distributions *f* parameterized by a parameter $\theta \in \Theta \subseteq R^{m}$, satisfying what we will call *generalized ϕ-heat equation*,
(24)$$\nabla_{\theta }f=K\;\mathrm{div}\left(∥\nabla_{x}\phi^{\prime }{\left(f\right)∥}_{\chi^{*}}^{\beta -2}\;\nabla_{x}f\right)$$
for some $K\in R^{m}$, possibly dependent on θ but not on *x*, and where ϕ is a convex twice differentiable function defined over a set $X\in R_{+}$.

When θ is scalar, this equation is an instance of what are known as quasilinear parabolic equations [119] (§ 8.8) and arises in various physical problems.

**Proposition** **5**(Extended de Bruijn identity)**.**
*Let f be a probability distribution with respect to a measure μ. Suppose that f is parameterized by a parameter $\theta \in \Theta \subseteq R^{m}$, and is defined over a set $X\subset R^{d}$. Assume that both X and μ do not depend on θ, and that f satisfies the nonlinear ϕ-heat equation Equation (Equation 24) for a twice differentiable convex function ϕ. Assume that $\nabla_{\theta }\phi \left(f\right)$ is absolutely integrable and locally integrable with respect to θ, and that the function ${∥\nabla_{x}\phi^{\prime }\left(f\right)∥}_{\chi^{*}}^{\beta -2}\;\nabla_{x}\phi \left(f\right)$ vanishes at the boundary of X. Thus, distribution f satisfies the extended de Bruijn identity, relating the ϕ-entropy of f and its nonparametric (β,ϕ)-Fisher information as follows,*
(25)$$\nabla_{\theta }H_{\phi }\left[f\right]=K\;C_{\phi }^{1-\beta }\;I_{\beta ,\phi }\left[f\right]$$
*with $C_{\phi }$ is the normalization constant given Equation (Equation 16).*

**Proof.** From the definition of the ϕ-entropy, the smoothness of the assumption enables to use the Leibnitz’ rule and differentiate under the integral,
$$\begin{array}{ccc} \nabla_{\theta }H_{\phi }\left[f\right] & = & -\int_{X}\phi^{\prime }\left(f\left(x\right)\right)\;\nabla_{\theta }f\left(x\right)\;d\mu \left(x\right)\\ & = & -K\int_{X}\phi^{\prime }\left(f\left(x\right)\right)\;\mathrm{div}\left(∥\nabla_{x}\phi^{\prime }{\left(f\left(x\right)\right)∥}_{\chi^{*}}^{\beta -2}\;\nabla_{x}f\left(x\right)\right)\;d\mu \left(x\right)\\ & = & -K\int_{X}\mathrm{div}\left(\phi^{\prime }\left(f\left(x\right)\right){∥\nabla_{x}\phi^{\prime }\left(f\left(x\right)\right)∥}_{\chi^{*}}^{\beta -2}\;\nabla_{x}f\left(x\right)\right)\;d\mu \left(x\right)\\ & & +K\int_{X}\nabla_{x}^{t}\phi^{\prime }\left(f\left(x\right)\right){∥\nabla_{x}\phi^{\prime }\left(f\left(x\right)\right)∥}_{\chi^{*}}^{\beta -2}\;\nabla_{x}f\left(x\right)\;d\mu \left(x\right)\\ & = & -K\int_{X}\mathrm{div}\left(∥\nabla_{x}\phi^{\prime }{\left(f\left(x\right)\right)∥}_{\chi^{*}}^{\beta -2}\;\nabla_{x}\phi \left(f\left(x\right)\right)\right)\;d\mu \left(x\right)\\ & & +K\int_{X}{\left(\phi^{\prime \prime }\left(f\left(x\right)\right)\right)}^{\beta -1}\;{∥\nabla_{x}f\left(x\right)∥}_{\chi^{*}}^{\beta }\;d\mu \left(x\right) \end{array}$$
where the second line comes from the ϕ-heat equation and where the third line comes from the product derivation rule.Now, from the divergence theorem, the first term of the right hand side reduces to the integral of $∥\nabla_{x}\phi^{\prime }{\left(f\right)∥}_{\chi^{*}}^{\beta -2}\;\nabla_{x}\phi \left(f\right)$ on the boundary of X, that vanishes from the assumption of the proposition, while the second term of the right hand side gives the right hand side of (Equation 25) from $C_{\phi }$ and the (β,ϕ)-Fisher information given by Equations (Equation 16) and (Equation 17) and Definition 6. □

As for the Cramér–Rao inequality, in the classical settings there exist matrix variants of the de Bruijn identity, the scalar form being a special one [115,117].

Coming back to the special examples we presented all along the paper:

**Example** **1** **(cont.).**
*In the Shannon entropy context, for $K=\frac{1}{2}$ and β=2, the standard heat equation is recovered and the usual de Bruijn identity is recovered.*


**Example** **2** **(cont.).**
*The case where $\phi \left(y\right)=y^{q}$ was intensively studied in [90] and the results of the paper are naturally recovered. In particular, the generalized ϕ-heat equation appears in anomalous diffusion in porous medium [90,119,120,121,122].*


**Example** **3** **(cont.).**
*For $\phi \left(x,y\right)=\phi \left(y\right)=\frac{1}{y}$, once again one finds the same form for the generalized heat equation than in [90,120,121], and therefore the same form of the generalized de Bruijn identity of [90] (see the items related to the escort distributions).*


## 6. Concluding Remarks

In this paper, we extended as far as possible the identities and inequalities which link the classical informational quantities—Shannon entropy, Fisher information, moments, etc., in the framework of the ϕ-entropies. Our first result concerns the inverse maximum entropy problem, starting with a probability distribution and constraints and searching for which entropy the distribution is the maximizer. If such a study was already tackled, it is extended here in a much more general context. We used general reference measures—not necessarily discrete or of Lebesgue. We also considered the case where the distribution and constraints do not share the same symmetries, which leads to state-dependent entropic functionals. Our second result is the generalization of the Cramér–Rao inequality in the same setting: to this end, a generalized Fisher information and generalized moments are introduced, both based on a convex function ϕ (and a so-called ϕ-escort distribution). The Cramér–Rao inequality is saturated precisely for the maximum ϕ-entropy distribution with the same moment constraints, linking all information quantities together. Finally, our third result is the statement of a generalized de Bruijn identity, linking the ϕ-entropy rate and the ϕ-Fisher information of a distribution satisfying an extended heat equation, called ϕ-heat equation.

As a direct perspective, the extensions of the generalized moments and Fisher information in terms of matrix, and matrix form of the generalized Cramér–Rao inequalities and de Bruijn identities are still open problems. Several ways to define matrix moments and Fisher information may be considered, such as in a term-wise manner as evoked in this paper. However, deriving matrix forms of the inequalities and identities does not seem trivial, and neither does obtaining the scalar form, for instance, through trace operator. Moreover, as the de Bruijn identity can be closely related to the generalized Price’s theorem [123,124,125], studying the connections between the extended de Bruijn and this theorem, or generalizing following the work of [125] is also of great interest.

Furthermore, two important inequalities are still lacking: The first one is the entropy power inequality (EPI), which states that the entropy power (exponential of twice the entropy) of the sum of two continuous independent random variables is higher than the sum of the individual entropy powers (In fact, there exist other equivalent versions which can be found, e.g., in [34,75,107,126,127,128].). The second one is the Stam inequality which lower bounds the product of the entropy power and the Fisher information. For the former, despite many efforts, the literature on extended version only considers special cases. For instance, some extensions in the classical settings exist for discrete variables but are somewhat limited [129,130,131]. In the continuous framework, the EPI was also extended to the class of the Rényi entropy (log of a ϕ-entropy with $\phi \left(u\right)=u^{\alpha }$) [132]. Note that variants of the EPI also exist in the context where one of the variables is Gaussian. This is equivalent to the convexity property of $\theta \mapsto N(X+\sqrt{\theta }Y)$ with *N* the entropy power and *Y* a Gaussian noise independent on *X* [133,134,135,136,137]; property also extended in the context of the Rényi entropy [132,138,139,140]. An important property that plays a key role in the inequality is the fact that the Rényi entropy is invariant to an affine transform of unit determinant and monotonic under convolution, a property which seems lost in the very general setting considered here. This fact leaves little room to extend the EPI in our general settings. Concerning the Stam inequality, at a first glance, the fact that the proof is based on the EPI seems to close any hope to extend it to the ϕ-entropy framework. However, it was remarkably extended to the Rényi entropy, based on the Gagliardo–Nirenberg inequality [84,86,87,141]. Nevertheless, a key property is that both the entropy power and the extended Fisher information have scaling properties that are lost in the general setting of the ϕ-entropies. A possible way to overcome the (apparent) limits just evoked could be to mimic alternative proofs such as those based on optimal transport [142]. This approach precisely drops off any use of Young or Sobolev-like inequalities. As far as we see, there is thus a little room for extensions in the settings of the paper. Both the extension of the EPI and the Stam inequality are left as perspectives.

Another perspective lies in the estimation of the generalized moments from data (or from estimates). Such a possibility would confer an operational role to our Cramér–Rao inequality, i.e., by computing the estimator’s generalized moments and comparing them to the bound. A difficulty resides in the presence of the ϕ-escort distribution which forbids empirical or Monte Carlo approaches. The escort distribution needs to be estimated. This problem seems not far from the estimation of entropies from data and plug-in approaches used in such problems can thus be considered, like kernel approaches [143,144,145], nearest neighbor approaches [145,146], or minimal spanning tree approaches [42]. Of course, this perspective goes far beyond the scope of this paper.

## Data Availability

Not applicable.

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
