# Peer review of "ϕ*-Informational Measures: Some Results and Interrelations"

_entropy, 2021, doi:10.3390/e23070911_

Round 1

Reviewer 1 Report

please attention at number of pages

Author Response

We are glad to see that the reviewer raised no specific issues to this work. We have taken into account the overall evaluation of the reviewer, improving as far as possible our motivation, descriptions, and presentation of the results. Our modifications follow also recommendations of the second and third reviewers. These modifications are thus detailed in the answers of reviewer 2 and reviewer 3. In addition, we asked for a bilingual colleague to have a deep reading of the English of the paper.

Reviewer 2 Report

please find my remarks in the attached pdf file.

Reviewer 3 Report

This is a fine paper with important consequences to the (growing) research area.

The authors are urged to further explore connections of these findings with the concavity (with added noise) of entropy power (Costa, 1985; Dembo, 1989; Villani, 2000, Toscani, 2014), and analogous relations in geometry related to the Brunn Minkowski inequality (Costa and Cover,1984).

Also of interest are the connections of these relations with Price's Theorem (Price, 1958; Pawula, 1967; Riba and de Cabrera, 2019).

Round 2

Reviewer 2 Report

The authors have addressed all my concerns and I have no further comments.